# Influence of Spirituality and Religiosity of Cancer Patients on Their Quality of Life

**DOI:** 10.3390/ijerph19094952

**Published:** 2022-04-19

**Authors:** Anna Majda, Natalia Szul, Kinga Kołodziej, Agata Wojcieszek, Zygmunt Pucko, Kinga Bakun

**Affiliations:** 1Faculty of Health Sciences, Institute of Nursing and Midwifery, Jagiellonian University Medical College, ul. Michałowskiego 12, 31-126 Kraków, Poland; anna.majda@uj.edu.pl (A.M.); agata.wojcieszek@uj.edu.pl (A.W.); zygmunt.pucko@uj.edu.pl (Z.P.); kinga.bakun@uj.edu.pl (K.B.); 2Maria Sklodowska-Curie Memorial Institute of Oncology, ul. Garncarska 11, 31-115 Kraków, Poland; szulnatalia97@gmail.com

**Keywords:** spirituality, religiosity, quality of life, cancer

## Abstract

Taking into account the spiritual and religious dimensions is important when it comes to taking care of patients with cancer and their quality of life. The study aimed to show the degree of dependence between the level of spirituality/religiosity of people who have just been diagnosed with cancer or have been diagnosed with cancer in the past and their quality of life. The Daily Spiritual Experience Scale (DSES), the EORTC QLQ-C30 quality of life and EORTC QLQ-FA12 fatigue-related quality of life questionnaire were used. One hundred one respondents of the Catholic faith obtained 65.22 points in DSES; 49.84 points on the QLQ-C30 functioning scale, 58.75 points on the physical scale, 60.73 points on the social scale, 50.17 points on the emotional scale, 64.69 points on the cognitive scale, 55.45 points in fulfilling one’s role and 28.38 points in financial impact. In the QLQ-FA12, respondents obtained 45.94 points on the physical scale, 47.53 points on the emotional scale and 30.69 points on the cognitive scale. In the respondents’ opinion, fatigue was the disease that reduced their quality of life the most—on average, 51.27 points. The oncological patients were characterized by a high level of spirituality/religiosity and an average level of quality of life. Spirituality/religiosity had a positive relationship with physical, emotional and social functioning. On the other hand, it was negatively associated with disease symptoms, such as pain or emotional and physical fatigue. Future research is needed in the context of the quality of life, focused on the spiritual and religious sphere of functioning of cancer patients, conducted in various cultural, ethnic and religious circles, which can serve to improve the education of nurses and develop their spiritual competences.

## 1. Introduction

Neoplastic diseases are currently one of the most common causes of death in Poland and rank right after cardiovascular diseases. The data of the Central Statistical Office show that the number of cancer cases and deaths is tending to increase [1]. This situation is a huge challenge for the Polish healthcare system. It also raises the bar of the native art of treatment, focusing its therapeutic action on an individual patient, because when the patient learns about the diagnosis of cancer, the disintegration of all dimensions of the patient’s existence begins. The patient quickly realizes that nothing will ever be the same again. The certainty of the disintegration of the current form of oneself and the change in lifestyle in the world raise a strong fear of lowering the quality of life and the possibility of gradual dying [2]. The result of traumatic experiences forces a person to ask fundamental questions dictated by the need to clarify existence. In the humanistic literature, these questions are called “damned questions”. They are usually placed within the limits of pure reason or within the horizon of faith. If they are formulated on the basis of cogito, they are existential and spiritual in nature. However, when they fall into the denominational field, they acquire a religious and spiritual value. At the same time, uttering them dreams only if they are limited to a confessional scheme in which God is understood in terms of a person who is at the same time the source of all good and the guardian of moral order. It is unfounded to pose “cursed questions” in the area of, for example, Buddhism or Taoism, because their theodicy does not contain a personified idea of God. Thus, the questions would fall into a metaphysical void.

Regardless of the context of the questions asked, however, they are always related to the deepest dimension of the human being, which is spirituality. As is known, in the scientific discourse, the issue of spirituality is not easy to explain or close in a strict definition. Methodologically, spirituality is divided into two types: secular and religious. Spirituality understood in the secular sense is the result of religion fatigue, especially in the monotheistic version, and is an alternative for all people who are looking for a different vision of reality, other content that creates human identity and explains the meaning of life [3]. This type of spirituality is not some kind of new religiosity rooted in the sphere of the sacred, but a state of mind, a means of self-expression and a type of interest focused on higher values or an undefined form of transcendence [4]. It can be found in almost every field of human activity, including medicine, ecology, science, economy, business and and politics. Everywhere, wherever it appears, it serves as a normative criterion necessary to evaluate things and phenomena [5]. However, despite its prevalence, it is difficult to measure [6,7].

Meanwhile, spirituality considered in connection with religion is intertwined with religiosity, which is expressed in the intensity of the believer’s involvement in religious practices, the subjective tightening of ties with the sacred and the faith cultivated by a given community [8]. The distinguishing features of this type of spirituality are its dependence on the institutional management of personal experiences and adherence to the rigors of canon law. In the ongoing debate, also in the field of the psychology of religion, there has long been talk of the difference between religiosity and spirituality with a religious profile [9]. However, many researchers consider such a distinction not convincing enough. According to Brian Zinnbauer and Kenneth Pargament, religiosity and religious spirituality are multidimensional phenomena and interpenetrate on different levels [10]. In turn, William Gavent and David Coulter suggest that this species of spirituality can be described in various ways without specifying its essence because as a phenomenon it goes beyond rigid divisions and precisely defined definition frames [11]. According to the above-mentioned authors, however, properly developed research tools make it possible to capture the constitutive features of religious spirituality [11]. According to this view, religious spirituality is a social phenomenon that manifests itself, inter alia, in collective rituals. The communal forms of rites are observable and therefore can be measurable [12,13].

From the perspective of religious spirituality, a pious person is therefore one who takes part in public ceremonies and observes religious norms [5,6,7]. If the analysis of the pattern of a pious person is limited to monotheistic religions, then it will be someone who, apart from following religious principles, entrusts his or her fate to God.

The above-mentioned varieties of spirituality can be cultivated for a long time. All too often they crystallize in a lifetime process. Both types of spirituality also have their own specific dynamics and depth. Traumatic events, such as cancer, can significantly contribute to changing the pace of spiritual development and the degree of its intensity. For example, some Christian patients, including Catholics, Orthodox, Protestants and Greek Catholics, in whom oncological treatment gave positive results claim that faith in God helped them strengthen their inner energies and heal [14]. This group of people has a strong sense of the purposefulness of their suffering and is convinced that it has been included in the eternal plan of the Creator [15]. They think about their illness, among other things, as a life lesson given to them by God, as a chance to smooth their own existence or as a punishment for moral trespasses [16]. This model of spirituality restores confidence in many cancer patients in the moral order of the world, reconciles with fate and alleviates suffering. In others, however, it causes regret and guilt, which sometimes generate anxiety and depression [17].

Researchers suggest that the involvement of cancer patients in the sphere of their religious spirituality is reflected in the frequency of their participation in liturgy, prayer meetings and the performance of various religious acts. For patients who profess Islam, such an act is to meditate on verses of the Koran or say a rosary consisting of 33 beads or three parts of 33 beads, helping to pronounce all the names of Allah. Patients who identify with the religion typical of the religious tradition of the Far East, mainly India, China and Japan, repeat mantras in their minds and choose the right pace of breathing for each of them. Oncological patients, on the other hand, who prefer different versions of nondenominational spirituality, find in their conceptual constructs contents that help them to correct life priorities, establish a new hierarchy of needs, integrate internally, discover other values, revive moral resistance, awaken hope, strengthen a sense of security, activate the ability to transgress one’s own borderline situation and influence the shape of existence [18,19]. The scope and strength of the interaction of these variables suggest that in the group of cancer patients, the spiritual/religious dimension is an integral part of the treatment course and has a direct impact on the quality of life [14].

Expert knowledge on the quality of life shows that it is a subjectively perceived matter, depending, among other things, on the attitudes, values and needs of a given person. The quality of life also evolves over time and changes with the disease. So, it is the sum of many components. The quality of human life can be defined on the basis of the assessment of its functioning in four spheres: physical, mental, social and spiritual/religious. In the physical area, the quality of life is determined by the severity of the symptoms of the disease and the possible side effects of treatment. In the psychological dimension, it is conditioned by, for example, a sense of loneliness and uncertainty. In the social component, the quality of life is greatly influenced by limitations related to fulfilling public roles. In turn, in the spiritual and religious space, an important factor may be, for example, the nature of the relationship to God or the relationship to some important idea [20].

Despite the lack of a paradigmatic definition of spirituality in nursing [21], attempts are being made to reach a consensus on the definition of spirituality. There has recently been agreement on the definition of spirituality in the context of palliative care: spirituality is a dynamic and internal aspect of humanity through which people seek ultimate meaning, purpose and transcendence and experience relationships with themselves, family, others, community, society, nature and signifier or saint. Spirituality is expressed through beliefs, values, traditions and practices [22].

Research abroad provides increasing insight into the definition and importance of spirituality in healthcare in general and nursing in particular. Beyond the question of definition, the importance of spirituality and spiritual care for nursing becomes clearer in practice-related research. The reviews showed that the quality of spiritual care is related to the specific needs of patients and the development of nurses’ competencies and the role of education [23,24,25]. McSherry and Ross say that research shows that nurses view spiritual care as a legitimate and fundamental aspect of nursing practice [26]. The analysis of Polish studies shows that they rarely combine spirituality or religiosity with the quality of life of, for example, cancer patients.

The aim of the study was to investigate the degree of influence of spirituality/religiosity on the quality of life of cancer patients. Determining this scale of impact will undoubtedly enrich the professionalism of healthcare professionals and improve holistic patient care by nurses. The study described in this article hypothesized that the Christian/Catholic spirituality model influences the perception and dimensions of the quality of life of cancer patients.

## 2. Materials and Methods

### 2.1. Research Design and Data Collection

The research was cross-sectional and was carried out using the method of diagnostic survey and estimation. The techniques that were used to collect the material for research were the survey technique and the scaling technique.

The survey consisted of one stage—the filling out of questionnaires by the respondents. The respondents’ informed consent for their participation in the study was obtained. The respondents were informed about the purpose of the study, keeping anonymity, voluntary participation in the study, the possibility of withdrawing from it at any time (without giving a reason) and the lack of consequences if they did not consent to participate.

The research was carried out in the departments of a hospital located in the Małopolskie Voivodeship and in a health clinic in the Podkarpackie Voivodeship. The research lasted from March to May 2021. One hundred fifty questionnaires were distributed, which the respondents meeting the criteria for inclusion in the study completed on their own or with the help of members of the research team. One hundred twenty-two research tools completed by respondents and 28 blank ones were obtained. The return rate was 81.33%. The authors believe that the reasons for not answering the questions were their sensitive, overly personal nature and the influence of the variable of social approval imbued in tools that measure feelings and moral behavior. In addition, the respondents’ awareness of the right to withdraw from participation in the study at any stage without giving a reason could have resulted in such a percentage of return. Additionally, 21 research tools were completed incorrectly (missing data, providing more than one answer to some questions in the QLQ and the Daily Spiritual Experience Scale). It can be assumed that this was dictated by the subjects’ lower cognitive and motor skills (fatigue, shaking hands). Ultimately, the number of people who correctly completed the research tools was 101.

### 2.2. Instruments

Sociodemographic data were obtained on the basis of a proprietary questionnaire. It consisted of questions about gender, age, place of residence, marital status, education, type of cancer and treatment, duration of treatment and faith/religion.

(a)Daily Spiritual Experience Scale (DSES)

The level of spirituality/religiosity was assessed using the Daily Spiritual Experience Scale (DSES) by the American researcher Lynn G. Underwood. The scale consists of sixteen questions. Fifteen of these are about daily spiritual experiences such as inner peace, delight, giving and receiving compassionate love, and gratitude for blessings received. The sixteenth question is general in nature and is about feeling close to God. The tool is universal in nature and does not depend on the religious beliefs of an individual [21,22]. The first 15 questions are assessed using a modified, six-point Likert scale, in which the answer categories are: “many times a day” (=1); “every day” (=2); “most days” (=3); “some days” (=4); “once in a while” (=5); and “never or almost never” (=6). Question 16 is graded using a modified Likert scale of four points, with answers: “not close” (=1); “somewhat close” (=2); “very close” (=3); and “as close as possible” (=4). The maximum possible number of points to be obtained by the respondent is 94 points, and the minimum is 16 points. The higher the test result, the more spiritual/religious he or she is [21,22]. In this study, the last question was rescaled and the following points were assigned to individual answers: “not close” (=1); “somewhat close” (=3); “very close” (=4); and “as close as possible” (=6), and the maximum sum of points to be obtained is 96. The relative accuracy of the scale measured by the Cronbach’s α coefficient, depending on the given population, is within the range of 0.86–0.96. The absolute validity of the test–retest method is r = 0.85. Internal consistency equal to 0.94 confirms that the scale is an internally valid measure. The factor analysis confirms its two-factor character for all the populations studied so far, including the population of Poland [27,28,29].

(b)Quality of Life of Cancer Patients (EORTC QLQ-C30)

In order to assess the quality of life, the EORTC QLQ-C30 questionnaire (version 3.0) created by the European Organization for Research and Treatment of Cancer was used. The questionnaire consists of thirty questions, which are grouped into 5 functional subscales, which include: physical functioning (5 questions), emotional functioning (4 questions), functioning in life roles (2 questions) and social functioning (2 questions). It also includes 3 symptom scales: fatigue (3 questions), nausea and vomiting (2 questions) and pain (2 questions). Additionally, it consists of 6 individual questions assessing the intensity of symptoms, such as insomnia, diarrhea, loss of appetite, dyspnea, constipation and financial difficulties. The last two questions are general and concern the assessment of the health condition and quality of life of the examined person. Responses from the first 28 questions in the questionnaire are graded using a modified, four-point Likert scale, in which the response categories are: “not at all” (=1); “a little” (=2); “quite a bit” (=3); and “very much” (=4). Questions 29 and 30 are assessed using a modified 7-point Likert scale, with grade 1 being “very poor” and grade 7 being “excellent”. The minimum possible score is 30, and the maximum is 126 points, but according to the theoretical assumption of the scale, the score is calculated in the range of 0–100 points. In questions 29 and 30, the greater the number of points obtained by the respondent, the better the functioning. For questions 1–28, the scoring is reversed; the higher the score, the greater the severity of persistent symptoms and the lower the quality of life. Reliability, validity and internal consistency of the EORTC QLQ-C30 scale were confirmed, also in research conducted in Poland [30,31,32]. 

(c)Quality of Life—Cancer-Related Fatigue (EORTC QLQ-FA12)

The Quality of Life—Cancer-Related Fatigue (EORTC QLQ-FA12) questionnaire developed by the European Organization for Research and Treatment of Cancer (EORTC) was used to assess the degree of fatigue associated with previous or ongoing cancer. The scale consists of twelve questions specifically related to fatigue: physical, emotional and cognitive. Responses are graded using a modified 4-point Likert scale, in which the response categories are: “not at all” (=1); “a little” (=2); “quite a bit” (=3); and “very much” (=4). The possible score range is 12–48. The higher the score, the greater the fatigue associated with cancer. In this study, the results were scaled to the range of 0–100 points. The reliability of the scale calculated using Cronbach’s α for the three subscales and the total score was 0.79–0.92 and 0.92, respectively, also in research conducted in Poland [27,33,34,35,36,37].

The tools are included in Appendix A.

### 2.3. Participants

The study group consisted of 101 people, including 56 women (55%) and 45 men (45%) who met the inclusion criteria for the study (Table 1).

The largest percentage of the respondents were people aged 60–70—26 people (25.7%). The second largest group were people aged 70–80—23 people (22.8%). Detailed statistics are presented in Table 2. Of the respondents, 45.5% lived in the countryside; the rest lived in cities, and 36.6% lived in cities with more than 100,000 inhabitants. Half of them lived in the Małopolskie and Podkarpackie voivodships. The largest groups were married people (57%) and people with secondary education (37.6%). The most frequently diagnosed neoplasms in the respondents were breast, lung, colon and prostate neoplasms. The respondents underwent surgery (71%), chemotherapy (54%) and radiotherapy (41%). Half of the respondents completed oncological treatment, which lasted from half a year to 5 years, and the other half were still undergoing oncological treatment. All respondents described themselves as Catholics.

### 2.4. Data Analysis

The study used statistical methods to enable the elaboration and interpretation of the results. Qualitative analysis was performed by calculating the number and percentage of occurrences for each variant. Nonparametric tests were used in the study. The comparison of the values of quantitative variables in two groups was performed using the Mann–Whitney U test. Correlations between quantitative variables were analyzed using the Spearman rank correlation coefficient. The Kruskal–Wallis test was used to compare groups for quantitative variables. Using the ANOVA Kruskal–Wallis test, the distributions of several variables were compared. A significance level of 0.05 was adopted in the statistical analysis. This means that all *p* values < 0.05 were interpreted as showing significant dependencies. The analysis was carried out using the R program (version 3.6.2) [38].

### 2.5. Ethical Considerations

Before the commencement of research activities, the consent of the directors of the institutions concerned was obtained. Ethical approval was obtained from the Academic Bioethics Committee (KBET/1072.6120.222.2020). The study was designed and conducted and its results were developed in accordance with the principles of (1) Good Scientific Practice, (2) the Helsinki Declaration and (3) Regulation (EU) 2016/679 of the European Parliament and of the Council of 27 April 2016 on the protection of natural persons with regard to the processing of personal data and on the free movement of such data, taking into account the (4) Repeal of Directive 95/46/EC (General Data Protection Regulation). The obtained data were anonymized, in consideration of (5) the Act of 10 May 2018 on the protection of personal data [39,40].

## 3. Results

The mean result regarding the respondents’ spirituality/religiosity was 65.22 points (SD = 21.05), and it was high. In terms of general health and quality of life, the respondents obtained an average of 49.84 points (SD = 25.60), and it was an average score. The examined people with neoplastic disease obtained an average score of 58.75 points (SD = 27.94) in terms of physical functioning; an average score of 55.45 points (SD = 33.50) in terms of performing their role; an average score of 60.73 points (SD = 36.02) in terms of social functioning; an average score of 50.17 points (SD = 30.86) in terms of emotional functioning, which was the lowest; and an average score of 64.69 points (SD = 29.65) in terms of cognitive functioning, which was the highest. The lowest impacts on the quality of life of the respondents were found for financial problems, 28.38 points (SD = 35.71), and symptoms such as lack of appetite, 32.34 points (SD = 35.10); dyspnea, 27.06 points (SD = 35.81); constipation, 26.07 points (SD = 28.52); nausea/vomiting, 13.53 points (SD = 24.91); and diarrhea, 9.90 points (SD = 21.88). The most significant impacts on the quality of life of the respondents were found for insomnia, 46.53 points (SD = 35.30), and pain, 45.71 points (SD = 34.93). The examined patients scored an average of 45.94 points (SD = 24.65) in terms of physical fatigue. The average score obtained for the respondents in terms of emotional fatigue was 47.53 points (SD = 35.83), which was the highest. People participating in the study scored an average of 30.69 points (SD = 30.16) in terms of cognitive fatigue, which was the lowest score. The detailed distribution of data by age groups in DSES, EORTC QLQ-C30 and EORTC QLQ-FA12 is presented in Table 3.

The results obtained for the respondents in terms of the level of spirituality/religiosity correlated significantly and positively with the results in the sphere of health and quality of life (r = 0.516; *p* < 0.001), physical functioning (r = 0.196; *p* < 0.048), fulfilling one’s role (r = 0.278; *p* < 0.004), emotional functioning (r = 0.312; *p* < 0.001) and social functioning (r = 0.351; *p* < 0.001). Accordingly, the DSES scores increased significantly with the increase in the assessment of the quality of life, which means that spirituality/religiosity had a positive impact on the quality of life of the cancer patients in such areas as health and quality of life; physical, emotional, and social functioning; and fulfilling one’s role. However, there was also a negative correlation between the level of spirituality/religiosity and fatigue (r = −0.316; *p* < 0.001), nausea (r = −0.240; *p* < 0.015), pain (r = −0.230, *p* < 0.020), dyspnea (r = −0.207; *p* < 0.037), insomnia (r = −0.261; *p* < 0.008), lack of appetite (r = −0.204; *p* < 0.040) and diarrhea (r = −0.204; *p* < 0.118). The obtained Spearman’s r coefficients for cognitive functioning, constipation and financial problems indicate no statistically significant correlations (Table 4).

The results obtained for the respondents in terms of the level of spirituality/religiosity correlated significantly and negatively with the results in the area of fatigue caused by cancer: physical fatigue (r = −0.397; *p* < 0.001) and emotional fatigue (r = −0.456; *p* < 0.001). There was no correlation between the level of spirituality/religiosity and cognitive fatigue (Table 5).

There was no difference in the level of spirituality/religiosity depending on the tumor location (*p* > 0.05) (Table 6).

Age correlated significantly (r = 0.226; *p* < 0.002) and positively (r > 0) with the level of spirituality and religiosity of the respondents. In addition, there was a difference in the level of spirituality/religiosity of the respondents depending on gender (z = 2.24; *p* = 0.02) and place of residence (*p* = 0.005); in the case of women, the median DSES score assumed a higher value, and among the respondents living in the city with over 100,000 of inhabitants, the median was lower. There was no difference in the level of spirituality/religiosity with regard to marital status (*p* = 0.055) and the level of education of the respondents (*p* = 0.297).

## 4. Discussion

As indicated in the introductory part of this article, exploring spirituality and religiosity is not an easy task. Researchers on this topic constantly face many barriers. A serious obstacle for them is, above all, the semantic inaccuracy of both concepts. Determining the correct correlation for these terms is also a big problem. However, specialists have been trying to define the boundaries and content of the actual equivalent for these names more accurately for a long time. The effect of their work is reflected not only in the field of scientific debate but also in the public space of some societies. For example, in the United States, a greater awareness of people has already been registered in the choice of words used to describe the phenomena of spirituality and religiosity. It is clearly demonstrated by the percentage of Americans who define themselves as spiritual but not religious, which increased from 19% to 27% between 2012 and 2017. On the other hand, research among Chinese people shows that the semantic differences between spirituality and religiosity are most clearly perceived by people who define themselves as nonreligious or atheists. This group of people is reluctant to engage in discussions related to religion but is ready to discuss topics related to spirituality [41].

The situation is slightly different when it comes to Poland. Although the terms religiosity and spirituality refer to two separate issues in foreign literature, they are often treated as synonyms of the same thing in Polish discourse. Therefore, their understanding and interpretation are different, not only due to the specificity of language, but also due to the established habits of thinking.

However, despite the difficulties mentioned above, the presented work provides a lot of important data of great cognitive value. The analysis of the results of studies conducted on a group of cancer patients of different ages from two voivodeships who underwent or still undergo oncological treatment shows that these people were characterized by a high level of spirituality/religiosity. This is due to several things. Firstly, the average result obtained was above the median (65.22 points). Second, a high percentage of respondents (38.6%) scored between 71 and 90 points. Thirdly, the respondents assessed their condition and quality of life as average (average score of 49.83 points).

Thanks to the conducted research, it is also known that the respondents made a convergent assessment, obtaining, on average, at least half of the possible points in the fields of physical, cognitive, emotional and social functioning. On the other hand, many ailments that may arise from the neoplastic process have been observed. The analysis of the respondents’ responses showed that nausea and vomiting, dyspnea, constipation and diarrhea were rather rare. The respondents slightly more often indicated the presence of fatigue, lack of appetite, pain and insomnia.

The statistical analysis showed the existence of a cause-and-effect relationship between the level of spirituality/religiosity and the level of quality of life as well as the level of fatigue caused by cancer. It has been proven that the higher the level of spirituality, the better the respondents assessed their health and quality of life. Similar results were obtained by Szatkowska et al., who studied the impact of the sense of inner coherence and bond with God on the quality of life. This impact showed that these two factors correlated with each other in terms of cognitive, emotional and social functions [42]. A similar influence of spirituality/religiosity on cancer patients is revealed in the research by Nowicki and his team. Based on their achievements, it can be argued that a higher level of faith is associated with a higher level of acceptance of the disease [43]. The cited effects of Nowicki et al. are consistent with the thesis of Żołnierz and Sak [12], who, after reviewing many studies in this field, noticed that they had one common denominator, that is, the positive influence of spirituality/religiosity on the quality of life in various dimensions of human existence.

However, slightly different results were obtained in the presented studies, which focused on somatic symptoms related to neoplastic diseases such as fatigue, nausea, dyspnea, insomnia, lack of appetite and diarrhea. The results obtained in *DSES* decreased with the increase in the assessment of the quality of life. Such a divergent result can be explained by the fact that better physical well-being of patients contributes to the extinction of the need to continue to entrust one’s health and life to God or a higher idea, e.g., fate. This would confirm the theory of Piskozub, according to which the increase in the level of spiritual/religious development usually takes place in particularly traumatic moments and is directly proportional to the scale of the threat to life [44].

In terms of sociodemographic variables, a positive correlation was noted between the level of spirituality/religiosity and the age of the respondents, which means that older respondents had higher levels of spirituality. At the same time, women with cancer were characterized by a higher level of spirituality/religiosity than men. On the other hand, a much lower level of religious belief was recorded among patients living in cities with more than 100,000 inhabitants. This is confirmed by the research results prepared by the Public Opinion Research Center in 2014, which showed that 70% of rural residents declare faith in God, compared to 55% of residents in the urban community being believers [45].

The influence of religiosity and spirituality on the quality of life of cancer patients is still a poorly understood phenomenon. Therefore, it is necessary to explore these issues further, improve research tools and identify new threads helpful in improving patients’ quality of life. Without further research, it is difficult even to imagine an accurate reading of the needs and commensurate satisfaction of patients’ expectations in the spiritual or religious area. The lack of a complete picture of the situation will not prevent unidentified elements from adversely affecting the quality of life, nor will it contribute to satisfaction with healthcare. Even worse, it can make treatment and nursing ineffective. Thus, maintaining the continuity of research, also in other religious groups, may be directly related to the daily practice of the art of treatment and nursing and shaping spiritual competencies among medical professionals, including nurses, and closely correlated with the quality of life of cancer patients.

### Limitations

The original research described in this article has two sides: the stronger and the weaker. The strength of the research carried out is recognizing the relationship between the declared spirituality/religiosity and the quality of life of the respondents with cancer, of the Christian/Catholic denomination. The study’s weakness was the result of factors such as a small study group, limiting the study to two places located in a religiously homogeneous area and the same geographical area, which was forced by the COVID-19 pandemic and the age of the respondents. Most of them were at the end of their lives. As is known, at this declining stage of human existence, religion/spirituality not only plays a significant role, but also has a specific form described as eschatological. It can be assumed that the results would be different if the cross-section of the respondents also included people from the earlier stages of ontogenesis and the intertwined developmental stages of spirituality/religiosity, such as the authentic stage or the period of stability. Another important disruptive factor could be sociodemographic variables such as age, gender and place of residence of the respondents. Most of them were in the late stages of life where religion/spirituality plays a significant role [46]. It can be assumed that the results would be different if the respondent’s cross-section included people from earlier stages of ontogenesis and men and women equally. Another disturbing sociodemographic may be the respondent’s perception of treatment effects, although no relationship was found between the quality of life and the type of cancer diagnosed in the respondents. It is noteworthy that although the percentage of respondents from large cities was higher, the level of their religiosity/spirituality was significantly lower, which may be related to the changing image of spirituality/religiosity of inhabitants from large agglomerations, also in the state of illness.

## 5. Conclusions

The spirituality/religiosity of Catholic people diagnosed with cancer positively influenced their quality of life in such areas as health; physical, emotional and social functioning; and fulfilling their role.

Further research focused on the spiritual sphere of functioning in the context of quality of life is needed, which will concern cancer patients and be conducted in various cultural, ethnic and religious contexts, in a larger number of recruited, age-diverse, inhabited areas.

Nursing education and management should take into account the emphasis on the development of spiritual competencies of Polish nurses in the context of working with oncological patients of various faiths.

## Figures and Tables

**Table 1 ijerph-19-04952-t001:** Study inclusion and exclusion criteria.

Inclusion Criteria for the Study	Criteria for Exclusion from the Study
- Gender: women and men - People suffering from cancer now or in the past - People professing a specific religion, and also nonbelievers - People without a diagnosed mental illness - Informed consent to participate in the study	People diagnosed with a mental illness Lack of informed consent to participate in the study

**Table 2 ijerph-19-04952-t002:** Sociodemographic data of the respondents.

Age (in Years)	n	%
<30	3	3
30–40	10	9.9
41–50	17	16.8
51–60	11	10.9
61–70	26	25.7
71–80	23	22.8
>80	6	5.9
**Gender**	**n**	**%**
Men	45	45
Women	56	55

n—number of people, %—percentage.

**Table 3 ijerph-19-04952-t003:** Distribution of respondents’ responses in terms of the level of spirituality/religiosity, quality of life and the level of fatigue caused by cancer.

Group										
Level of Spirituality/Religiosity (DSES)	Points									
	0–30		31–50		51–70		71–90		91–100	
	n	%	n	%	n	%	n	%	n	%
Level of spirituality/religiosity	8	7.9	17	16.8	26	25.7	39	38.6	11	10.9
**Group**										
**Quality of life (EORTC QLQ-C30)**	**Points**									
	**0–20**		**21–40**		**41–60**		**61–80**		**81–100**	
	**N**	**%**	**N**	**%**	**N**	**%**	**N**	**%**	**N**	**%**
General health and quality of life	15	14.8	21	20.8	30	29.7	19	18.8	16	15.9
Physical functioning	10	9.9	21	20.8	19	18.8	30	29.7	21	20.8
Role functioning	18	17.8	19	18.8	16	15.9	16	15.9	32	31.6
Emotional functioning	22	21.8	13	12.9	26	25.7	16	15.9	24	23.7
Cognitive functioning	16	15.9	5	4.9	12	11.9	27	26.7	41	40.6
Social functioning	21	20.8	13	12.9	6	5.9	19	18.8	42	41.6
Fatigue	7	6.9	1	30.7	28	27.7	22	21.8	13	12.9
Nausea and vomiting	80	79.2	8	7.95	3	2.95	6	5.95	4	3.95
Pain	33	32.7	19	18.8	13	12.9	8	7.9	28	27.7
Dyspnea	54	53.5	24	23.7	0	0	10	9.9	13	12.9
Insomnia	24	23.7	33	32.6	0	0	24	23.7	20	19.8
Appetite loss	45	44.6	26	25.7	0	0	18	17.8	12	11.9
Constipation	45	44.6	38	37.6	0	0	13	12.9	5	4.9
Diarrhea	79	78.22	17	16.8	0	0	2	1.98	3	2.97
Financial difficulties	52	51.5	26	25.7	0	0	9	8.9	14	13.9
**Group**										
**Quality of Life—Cancer-Related Fatigue (EORTC QLQ-FA12)**	**Points**									
	**0–20**		**21–40**		**41–60**		**61–80**		**81–100**	
	**N**	**%**	**N**	**%**	**N**	**%**	**N**	**%**	**N**	**%**
Physical fatigue	17	16.8	35	34.7	23	22.8	15	14.8	11	10.9
Emotional fatigue	23	22.8	27	26.7	11	10.9	15	14.85	25	24.75
Cognitive fatigue	51	50.5	19	18.8	12	11.9	10	9.9	9	8.9

N—number of people, %—percentage.

**Table 4 ijerph-19-04952-t004:** Relationship between the perception of the quality of life and the level of spirituality/religiosity of cancer patients.

Quality of Life (EORTC QLQ-C30)	Correlation with the Level of Spirituality/ Religiosity (Points)
	Spearman’s Correlation Coefficient
General health and quality of life	r = 0.516, *p* < 0.001
Physical functioning	r = 0.196, *p* < 0.048
Role functioning	r = 0.278, *p* < 0.004
Emotional functioning	r = 0.312, *p* < 0.001
Cognitive functioning	r = −0.012, *p* = 0.904
Social functioning	r = 0.351, *p* < 0.001
Fatigue	r = −0.316, *p* < 0.001
Nausea and vomiting	r = −0.240, *p* < 0.015
Pain	r = −0.230, *p* < 0.020
Dyspnea	r = −0.207, *p* < 0.037
Insomnia	r = −0.261, *p* < 0.008
Appetite loss	r = −0.204, *p* < 0.040
Constipation	r = −0.011, *p* < 0.911
Diarrhea	r = −0.204, *p* < 0.039
Financial difficulties	r = −0.156, *p* < 0.118

*p*—significance level, r—Spearman’s correlation coefficient.

**Table 5 ijerph-19-04952-t005:** The relationship between the level of physical, emotional and cognitive fatigue caused by cancer and the level of spirituality/religiosity of the respondents.

Quality of Life—Cancer-Related Fatigue (EORTC QLQ-FA12)	Correlation with the Level of Spirituality/Religiosity (Points)
	Spearman’s Correlation Coefficient
Physical fatigue	r = −0.397, *p* < 0.001
Emotional fatigue	r = −0.456, *p* < 0.001
Cognitive fatigue	r = −0.145, *p* < 0.145

*p*—significance level, r—Spearman’s correlation coefficient.

**Table 6 ijerph-19-04952-t006:** Identification of differences in the level of spirituality/religiosity of the respondents depending on the location of the cancer.

The Type of Tumor	Level of Spirituality/Religiosity			
	N	Sum of Ranks	Average Rank	*p*
Genitourinary system	33	1975.50	59.86	*p* = 0.38 *
Digestive system	27	1243	46.03	
Respiratory system	15	566.5	37.76	
Skin	6	324	54	
Endocrine system	4	219.5	54.87	
Circulatory system	3	178.50	59.5	
Another type	8	376.50	47.06	
More than one organ	5	267,50	53.5	

* ANOVA Kruskal–Wallis test.

## Data Availability

Data sharing not applicable.

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
