# Peer review of "Influence of Spirituality and Religiosity of Cancer Patients on Their Quality of Life"

_ijerph, 2022, doi:10.3390/ijerph19094952_

Round 1

Reviewer 1 Report

Majda A et al investigated the relationship between the level of spirituality / religiosity of people who were diagnosed with cancer at present or in the past and their quality of life. Although they presented their methodology fair enough, unfortunately the manuscript lacks of novelty and scientific interest. There are major issues that need to be fixed including the structure of introduction and the part of discussion with the limitations of the study. However, they did a good effort in the interpretation of their results. It would be helpful to include your questionnaires as supplementary material.

Reviewer 2 Report

The manuscript submitted by Majda et al., titled: "Spirituality and religiosity of cancer patients towards their quality of life" is aiming at investigating the level at which spirituality/religiosity is affecting the quality of life of cancer patients. The topic is of interest with potential significant clinical implications since quality of life constitutes a key target for cancer patients. 

The paper is well structured and organized. The reviewer would like to suggest a read-through by an English native speaker for language improvement as it pertains to grammar, syntax, flow and optimal use/selection of vocabulary.

In addition the reviewer would like to offer the following points for improvement of the manuscript:

  1. Consider including a hypothesis at the end of the introduction section.
  2. In terms of spirituality did the authors consider the literature on other religions or denominations  (eg: Christian Orthodoxy versus Catholicism, and Protestantism, Buddhism, Islam?). 
  3. In terms of spirituality and moral sciences and practices did the authors investigate the literature for such aspects such as Freemasonry for example?
  4. Were the questionnaires used in the study validated for the particular population?
  5. What were potential confounding factors for the study? How were identified and addressed? (eg: Socioeconomic status, diet, different medication regime, etc).
  6. Consider the point made above (no. 2) in the discussion as well.

Reviewer 3 Report

The authors focus on a very important group of people and explore how the non-medical dimension can affect them. It’s clear that this study is significant, and even in some cases helps to alleviate anxiety and restlessness in this group, and helps to improve their quality of life. The purpose of the study is clear, and the method is appropriate. The manuscript meets the Journal requirement. Before it could be published, the following comments are for the author’s reference and some of them should be explained or clarified.

  1. The author mentions: 49 questionnaires were rejected due to incorrect or incomplete filling (Line 152~153). Why do so many questionnaires not meet the requirements? Is it because the questionnaire design may not be very reasonable? Or is it because of the objective factors of the respondents? If so, is it necessary to increase the number of respondents? It is suggested that the author could explain this.
  2. In Section 3, the author lists almost all the data very carefully (roughly 3 pages). However, if these statements are just another description of tables (Table 3, 4, 5, and 6), then we suggest that the author can take a more concise approach. Otherwise, such lengthy content, coupled with the reader’s need to switch pages repeatedly to compare “paragraphs” and “tables”, is not a good way to present. Or rather, the author only needs to list the core parts, and the rest is actually enough through the tables.
  3. Following the previous question, since the author sorted out the research results in great detail, we look forward to what kind of discussion and analysis the author can carry out. But, the discussion and final conclusions do not seem to be in-depth enough. For example, we are not saying that others’ findings cannot be quoted at this stage, but these quotations need to be used with caution. We are looking forward to seeing the author’s own point of view. In addition, the authors present the limitations of the study in this link, which should be placed at the end.
  4. The part of the conclusion, if possible, would be hoped to be further strengthened. Or, based on these 5 conclusions, are they effective in responding to the original purpose of the study? And what else needs to be paid attention to in the follow-up work?
  5. Minor issues with formatting and details need to be addressed through proofreading.
  • Abstracts must not exceed 200 words. The author needs to remove some of the content.
  • The formatting of some paragraphs needs to be checked again, there are some abnormal line breaks or unnecessary blanks. Such as: Line 226~227, Line 386~387, Line 406, and Line 481.
  • The author lists 38 references, but the numbers in the article are 1-37. Please check again by the author. In addition, there are some minor problems with the format of individual references, and the author can check them one by one according to the template of the journal.

Round 2

Reviewer 1 Report

Authors improved the manuscript and, thus, it could be published in its current form.

Author Response

Thank you for accepting the amendments made.

Reviewer 2 Report

The authors have made a reasonable effort to address reviewers comments. Proofreading is recommended.

Author Response

Thank you for appreciating our work in preparing the amendments. The authors made linguistic and stylistic corrections in the text of the manuscript - marked in green.

Reviewer 3 Report

It is obvious that the authors have put a lot of effort into reorganizing the structure of the article and further revising or refining the content of the manuscript. The authors respond clearly to the previous problems, and also moderately condense the content that is too lengthy.

There are minor in this manuscript that deserves further refinement as the followings:

  1. The limitations of the manuscript  may be better placed at the end. (Line 615)
  2. The current findings are more like a plan for follow-up studies, and it is recommended that the authors again summarize more definitive conclusions based on the findings of the study.
  3. We still recommend that authors check the cited references to determine if they must be cited here. Indeed, the author’s point of view must be reflected in the discussion part.
